# MimicTouch: Leveraging Multi-modal Human Tactile Demonstrations for Contact-rich Manipulation

**Kelin Yu**[*,1], **Yunhai Han**[*,1], **Qixian Wang**[1,2], **Vaibhav Saxena**[1], **Danfei Xu**[1], **Ye Zhao**[1]

[*]Equal Contribution
[1]Georgia Institute of Technology
[2]Zhejiang Technology
`kyu85, yhan389, qxian, vsaxena33, danfei, yezhao@gatech.edu`

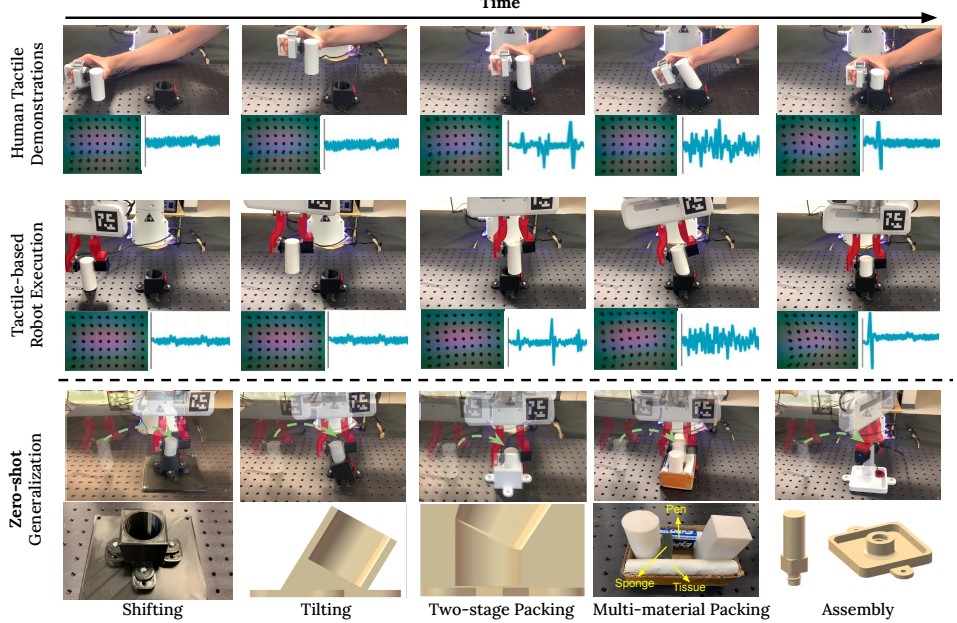

Figure 1: The first row shows the human tactile demonstrations, including the tactile and proprioception data. The second row shows the robot execution with tactile feedback. The third row below the dashed line describes the policy's **zero-shot** generalization capability in five different domains, including variations in hole positions, angles, inner shapes, materials, and a different assembly task.

**Abstract:** Tactile sensing is critical to fine-grained, contact-rich manipulation tasks, such as insertion and assembly. Prior research has shown the possibility of learning tactile-guided policy from teleoperated demonstration data. However, to provide the demonstration, human demonstrators often rely on visual feedback to control the robot. This creates a gap between the sensing modality used for controlling the robot (visual) and the modality of interest (tactile). To bridge this gap, we introduce "MimicTouch", a novel framework for learning policies directly from demonstrations provided by human users with their hands. The key innovations are i) a human tactile data collection system which collects multi-modal tactile dataset for learning human's tactile-guided control strategy, ii) an imitation learning-based framework for learning human's tactile-guided control strategy through such data, and iii) an online residual RL framework to bridge the embodiment gap between the human hand and the robot gripper. Through comprehensive experiments, we highlight the efficacy of utilizing human's tactile-guided control strategy to resolve contact-rich manipulation tasks. The project website is at `https://sites.google.com/view/MimicTouch`.

**Keywords:** Tactile Sensing, Learning from Human, Imitation Learning

8th Conference on Robot Learning (CoRL 2024), Munich, Germany.

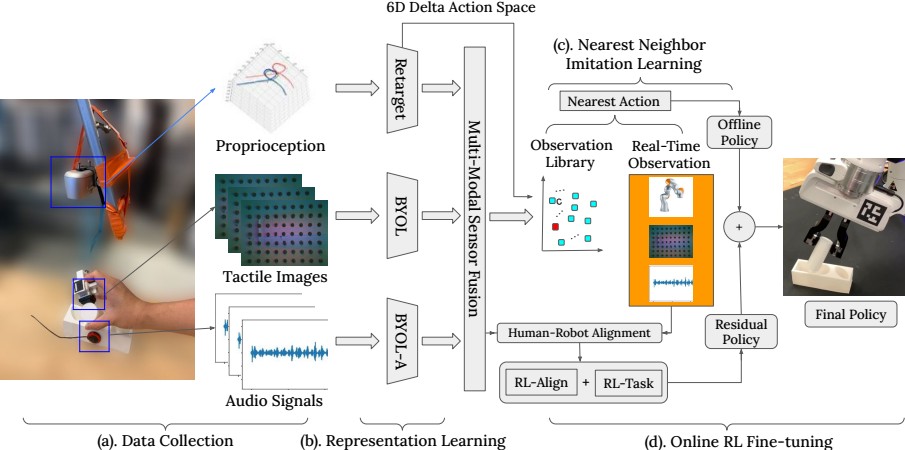

Figure 2: Illustration of the MimicTouch Framework. In part (a), we collect the multi-modal human tactile demonstrations. In part (b), we learn compact low-dimensional tactile representations. In part (c), we derive an offline policy through a non-parametric imitation learning method. In part (d), we refine the offline policy through online residual reinforcement learning on a physical robot.

# 1 Introduction

Enabling robots to perform contact-rich tasks such as insertion remains a formidable challenge in robotics. The primary reason is the complex dynamic interaction between the robot and the object which is influenced by various factors including intricate material properties and low tolerance for error. This necessitates an adaptive, data-centric insertion mechanism that utilizes real-time sensor feedback. Recent methods have heavily explored vision-based solutions to tackle this problem [1–4]. Notably, NVIDIA's sim-to-real transfer approach [1] achieves success rates of up to 99.2% in transferring assembly tasks using their customized "Factory" simulator [2]. However, vision-based approaches can fall short when visual feedback is compromised by cluttered occlusions or bad lighting conditions.

Humans exhibit fine-grained manipulation skills through tactile sensing, which allows for successful insertions by solely using tactile feedback to generate complex, continuous, and precise motions [5]. Motivated by this, recent studies collect demonstrations that combine various sensory inputs for policy learning [6–8]. However, these methods assume a limited action space, e.g., only 3D translations, to compensate for the complexity of collecting dynamic demonstrations. They also heavily rely on robot teleoperation systems [9–11], which inevitably creates a gap between the visual sensing used for data collection and the recorded tactile sensing for policy learning. As a result, these methods are unable to emulate human's tactile-guided control strategy for contact-rich tasks.

To tackle these challenges, we present "MimicTouch" (shown in Fig. 2), a novel framework that enables robots to learn human's tactile-guided control strategy. Specifically, MimicTouch first introduces a human tactile-guided data collection system to gather multi-modal tactile feedback (tactile + audio) directly from human demonstrators. Next, it incorporates a representation learning model to capture task-specific sensor input features. These compact representations enhance the performance of subsequent imitation learning by abstracting essential sensory information. Then, it employs a non-parametric imitation learning method [12] to derive an offline policy from the collected human tactile demonstrations. Finally, it leverages online residual reinforcement learning (RL) to fine-tune the offline policy on the physical robot, aiming to bridge the embodiment gap between the human hand and the robot gripper and enrich contact reasoning.

We conduct comprehensive experiments on contact-rich insertion tasks to evaluate the offline policy derived from demonstrations and the final policy fine-tuned via RL. We find that MimicTouch can collect tactile demonstrations more efficiently than teleoperation. More importantly, the MimicTouch policy can be effectively learned from such demonstrations and significantly outperforms the policy learned from teleoperation data. Additionally, we set up seven generalization tasks in five different domains and show the final policy exhibits superior **zero-shot** generalization capability.

## 2 Related Works

**Multi-modal tactile sensing.** Vision-based tactile sensing is integral to robotic manipulation as it excels at estimating local contact geometry and frictional properties [13–16]. It further enables imitation learning-based methods through policy observations [6, 7, 17] or reinforcement learning methods through reward signals [18–20] for various contact-rich manipulation tasks. Additionally, it is also widely used in shape reconstruction [21–23] and grasping [24–27]. On the other hand, audio-based tactile sensors, such as contact microphones have also been demonstrated effective in robotics applications such as manipulation [28], classification [29, 30], and dynamics modelling [31, 32]. These sensors can emulate nerve endings within human skin to better detect vibrations during tactile interactions. Therefore, incorporating both sensor modalities can yield tactile feedback more akin to human sensations, enabling the robot to learn a human-like tactile-guided control strategy.

**Learning from human demonstrations.** Learning from human demonstrations is a long-standing research topic. One group of methods learns the robot behaviors from human videos [33–35]. However, these methods adopt only visual sensing, which can be easily collected by low-cost cameras. While visual sensing can provide high-level scene understanding, contact-rich tasks require tactile feedback for precise execution. To incorporate the tactile data into the demonstrations, recent works use robot teleoperation systems [6–8, 36–38], where human uses teleoperation system to control a sensor-equipped robot during data collection. However, human operators must guide the robot using visual feedback, thereby creating a gap between the visual sensing used for data collection and the tactile sensing recorded for policy learning. Furthermore, collecting $6D$ dynamic motions for contact-rich manipulations via teleoperation is challenging. Therefore, in this work, we propose to collect human tactile demonstrations, in which the sensing gap is addressed and the demonstration motions are more versatile.

**Imitation learning.** Offline imitation learning (IL) is an effective strategy to learn robot policies in the real world. We consider two classes of IL methods: parametric methods [39–41] and non-parametric methods [12, 42, 43]. Parametric methods typically train neural networks to map observations to expert actions. While general in principle, they are prone to covariant shift and compounding errors [44]. Our method instead adopts a non-parametric imitation learning method. These methods constrain robot behaviors to the demonstrated data via techniques such as nearest-neighbor lookup [12]. While they may be less general, they offer a safer alternative to their parametric counterparts, which is crucial for the real-world contact-rich manipulation tasks considered in this work.

## 3 MimicTouch Framework

We aim to enable the robot to resolve contact-rich manipulation tasks by learning tactile-guided control strategy from human demonstrations. To achieve this, we propose a novel learning framework named "MimicTouch". It first introduces human tactile demonstrations (Sec. 3.1) collected by human hands. Then, to emulate the human's tactile-guided control strategy for successful robot execution, MimicTouch has three learning phases. Firstly, it learns lower dimensional tactile representations from the human tactile demonstrations in a self-supervised manner (Sec. 3.2). Next, it derives an offline policy with the learned representations using a non-parametric imitation learning method [12] (Sec. 3.3). Lastly, it refines the offline policy through online residual RL on the real robot(Sec. 3.4). The overall MimicTouch framework is shown in Fig. 2.

### 3.1 Collecting Human Tactile Demonstrations

To collect tactile demonstrations, current teleoperation systems have three key limitations: i). limited scalability due to the need for a robot to collect demonstrations [10], ii). long training time and expertise to become proficient with the system for fine-grained manipulation, and iii). sensing gap between the visual sensing used for collection and recorded tactile sensing. To address these, our key innovation is a system that enables humans to provide tactile demonstrations with their hands. The system is elaborated in Fig. 6 (Appendix. A), and it collects the pose of human fingertips, tactile images, and audio signals when human demonstrators perform contact-rich insertion tasks.

The data collection system consists of the following components. We use the RealSense camera with Aruco Marker [45] for human fingertip pose tracking. The tracked fingertip poses are then

treated as the robot end-effector's poses after calibration and filtering (Appendix. C.2). We also use the GelSight Mini [46], a compact vision-based tactile sensor that is conveniently mounted onto human fingertips using a custom fixture, to estimate the contacts between the object and the fingertip. Notably, we only use one tactile sensor in our experiment setup instead of two. The Audio data, which is helpful for manipulation tasks due to its sensitivity to contact vibration signals [28, 47], is captured using the HOYUJI TD-11 piezo-electric contact microphone. Considering the potential discrepancies in the mechanical vibrations between the human and the robot, the microphone is placed at the base of the insertion hole to ensure signal consistency.

## 3.2 Learning Tactile Representation

The policy learning on high-dimensional sensor inputs struggles with real-world deployment due to computational burden and sensor noise. Additionally, in this work, variations may arise between sensor inputs from human tactile demonstrations and real-time robot feedback due to different finger-object contact force. Inspired by recent works that learn lower-dimensional embeddings for imitation learning [6, 12, 48], we learn the compact representation for both tactile and audio data using self-supervised learning methods (part (b) in Fig. 2). Intuitively, it identifies a low-dimensional embedding space for different augmented data (tactile images or audio spectrum) and projects them to a similar embedding. As a result, these embeddings are more computationally efficient and more robust to task-irrelevant sensor noise. This learning phase consists of the following two parts:

**Data collection.** We collect task-specific tactile-audio data from the human demonstrator. The dataset encompasses successful, failed, and sub-optimal demonstrations. For each, we segment the audio data at 2Hz. In total, we collect 100 demonstration trajectories, including 7657 tactile images and 1,000 audio segments. More details are shown in Appendix. B.

**Self-supervised learning.** We employ the Bootstrap Your Own Latent (BYOL) [49] for tactile images and BYOL for audio (BYOL-A) for audio segments [50], since they have demonstrated desired performance in computer vision [49], audio learning [50], and robotics [7, 28, 36]. For BYOL and BYOL-A, the inputs are tactile images and audio spectrums separately, and the embedding spaces are both configured to be 2048-dimensional. Learning details are included in the Appendix. B.

## 3.3 Learning Offline Policy from Human Tactile Demonstrations

The next step is learning robot policy from the human tactile demonstrations. Here, one unique challenge is that the human hand moves much faster than the robot, resulting in sparse temporal observation-action samples (i.e., large action values per observation). Also, the embodiment gap (e.g., different motion capabilities and contact forces) can lead to out-of-domain demonstrations, which will make parametric policies prone to covariant shift [44] (see Sec. 4.2.1 for experimental validation). As a result, we use a non-parametric imitation learning method [12] to ensure the execution efficacy of the policy learned from human tactile demonstrations (part (c) in Fig. 2). In addition to the learning algorithm, data pre-processing is necessary for synchronization and the details are included in Appendix. C.

**Non-parametric imitation learning.** We build our algorithm on the NN-based framework [12], extending it to include tactile-audio representation without visual signals. At the $i$-th time step, the observations and actions are denoted as $(o_i^T, o_i^A, o^{EE}, a_i)$, where $o^T$ is the tactile representation, $o^A$ is the audio representation, $o^{EE}$ is the robot end-effector pose, and the action $a$ is defined by the $6D$ delta pose of the robot end-effector. Then, we extract tactile and audio features $(y_i^T, y_i^A)$ from $(o_i^T, o_i^A)$ using the pre-trained representation encoders, respectively. These tactile embeddings and the robot end-effector pose $(y_i^T, y_i^A, o_i^{EE})$ are formulated as the key features of the demonstration library. Given the varying scales of these inputs, we normalize them such that the maximum distance for each input is unity in the library. In robot execution, for a given real-time observation $(\hat{o}_i^T, \hat{o}_i^A, \hat{o}_i^{EE})$, we first obtain the query feature $(\hat{y}_t^T, \hat{y}_t^A, \hat{o}_t^{EE})$, and then search the demonstration library for a nearest-neighbor-based action prediction.

## 3.4 Learning Residual Policy through Online Reinforcement Learning

The offline policy learned from human tactile demonstrations might not guarantee task success when deployed on the physical robot. This could be due to: i). morphological differences between the

human hand and the robot gripper, ii). inaccurate fingertip tracking caused by fast movements, and iii). underexplored contact effects. Therefore, motivated by recent works using pure RL [17–20, 51] to learn tactile policies, we further leverage online reinforcement learning that allows in-domain robot interactions (part (d) of Fig. 2). It is noteworthy that the previous pure RL methods often generate quasi-static motions and utilize a limited action space [17–19, 51] because they learn from scratch without effective motion priors. On the contrary, we intend to leverage the best of both advantages by RL fine-tuning the offline policy learned from human tactile demonstrations.

Since it is infeasible to directly fine-tune the non-parametric policy, we instead learn a residual policy. The input to the residual policy $\pi_r$ consists of the current observation $(\hat{y}_i^T, \hat{y}_i^A, \hat{o}_i^{EE})$ the last policy output, which is the sum of the offline policy output and the residual policy output $a_{i-1}^o + a_{i-1}^r$. Considering we use $6D$ continuous action space and sparse observation-action pairs (around 70 actions per trajectory), we opt for SAC [52] to handle the continuous action space with entropy regularization and to generate a replay buffer to increase the size of training data. Finally, the robot action is the su, of the offline policy output and the residual policy output. To ensure that the residual policy only makes slight adjustments, we set specific action limits for the residual policy.

Another critical component of residual RL is the reward design, which must balance exploitation and exploration. To address this, we combine an expert-aligned reward, which is measured by the KL divergence between the human expert trajectory and the robot executed trajectory, with a task-specific reward. The expert-aligned reward encourages a policy distribution that mimics the demonstrations, whereas the task-specific reward drives exploration to optimize the in-domain robot policy. More details about pseudocode, action and policy design, and rewards are included in Appendix. D.

## 4 Experiments

In this section, we first describe the experiment setting and the data collection throughput to highlight that MimicTouch can efficiently collect useful demonstrations (Sec. 4.1). Then we introduce the Offline Policy Evaluation to validate the efficacy of the offline policy and highlight the benefits of using human tactile demonstrations (Sec. 4.2), and the Online Policy Improvement and Generalization Evaluation to demonstrate the efficiency of learning the residual policy through online RL and the superior zero-shot generalization capability (Sec. 4.3).

**Hardware setting.** All experiments are conducted on a Franka Emika Panda Arm. For each task, the learned policy generates the 6-DoF pose command and then maps it to 7-DoF joint torque actions using an inverse kinematics solver and a low-level built-in controller.

**Teleoperation setting.** We compare our human tactile-guided data collection system (Sec. 3.1) with Spacemouse-based teleoperation, a commonly used interface for manipulation tasks [9, 11, 37], and a Hand-guided teleoperation similar to Kinesthetic teaching (See Appendix. E for details). For both, to collect a similar number of observation-action pairs for each trajectory, we collect one robot state, one tactile image, and 0.5s audio segment at 5Hz. Since Spacemouse-based teleoperation requires considerable expertise, we allocate approximately 5 hours for practice with this system.

**Tasks.** We focus on insertion tasks that exemplify the challenge of many contact-rich manipulation tasks. We 3D-print a cylinder and an insertion hole base and set up the same task environments for both data collection settings. An example has been shown in Fig. 1, and the dimensions of each piece, including those used in the generalization tasks (Sec. 4.3), are detailed in Appendix. F.

### 4.1 Human Tactile Demonstration Collection System

In this section, we demonstrate that Human Tactile Demonstrations can greatly improve data collection throughput for contact-rich manipulation tasks. We begin by determining the **usability** of a demonstration trajectory based on the following two metrics: i) the object is successfully inserted into the hole without any slipping or falling, and ii) the task is completed within 100 actions. Using these criteria, we record the time length of collecting 20 usable demonstration trajectories by using our customized system, the Spacemouse-based teleoperation and the Hand-guided teleoperation. Then, we evaluate the data collection throughput (see Table. 1) in two metrics: i) the number of usable demonstrations collected per hour, and ii) the success rate for collecting usable demonstrations.

| Methods | Frequency | Success Rate |
|---|---|---|
| Spacemouse Teleoperation | 19 traj/hr | 38.5% (20/52) |
| Hand Teleoperation | 44 traj/hr | 58.8% (20/34) |
| Human Tactile Demonstrations | **104 traj/hr** | **83.3% (20/24)** |

Table 1: Data collection throughput for Human Tactile Demonstrations and Teleoperation systems.

The results in Table 1 support our insights: i) human tactile demonstrations can be collected significantly more efficiently than teleoperation systems for contact-rich tasks, and ii) human tactile demonstrations can seamlessly integrate human's tactile feedback and motion capability, whereas teleoperation systems struggle to capture such dynamic tactile-guided motions. These factors together result in much lower data collection efficiency and success rates of the teleoperation systems.

## 4.2 Offline Policy Evaluation

### 4.2.1 MimicTouch effectively learns from human tactile demonstrations

In this subsection, we evaluate offline policies learned from human tactile demonstrations using both a nearest-neighbor and a parametric imitation learning method. Our goal is to test if these policies can perform within the desired error tolerance in real-world settings, which is crucial for physical robot execution. To evaluate it, we use our data collection system (Sec. 3.1) to collect 20 noise-free data sequences as the datasets to learn both offline policies. For testing, we gather another 5 data sequences with random noise to emulate the real-world environments (details are in Appendix. G), and compute the mean square error (MSE, defined in Appendix. H) on the testing set. Also, we conduct the real-world experiments and compare the task success rate over 25 policy rollouts.

For the baseline parametric imitation learning method, we select the MULSA [6], which has been previously demonstrated effective in multisensory robot learning for insertion tasks. In our setting, we use the same sensor input $(y_i^T, y_i^A, o_i^{EE})$ as in the NN-based method to generate the continuous $6D$ delta action $a_i$. The same MSE loss is used for policy training and validation.

For both offline policies, we calculate the MSE losses on the testing set. We observed that the MSE loss from the NN-based policy is **0.21**, which is significantly lower than that of MULSA policy (**1.53**). Similarly, the NN-based policy achieves a **40% (10/25)** task success rate, while the MULSA policy achieves only **16% (4/25)**. We further provide three failure rollouts of the MULSA policy in Appendix I, to illustrate the reasons for its poorer performance. These results together suggest that the NN-based policy can generate more reliable $6D$ continuous actions, indicating it is more suitable for subsequent online real-world RL fine-tuning.

Additionally, we conduct ablation study on different sensor inputs in Appendix. J. The results suggest that multi-modal tactile feedback is crucial for the success of contact-rich insertion tasks.

### 4.2.2 Human tactile demonstrations trains better policies than teleoperated demonstrations

In this subsection, we compare the performance of the offline policies trained from human tactile demonstrations and teleoperation demonstrations. We collect 20 trajectories for each, and then we use the same NN-based method to learn the offline policies for each set of demonstrations. We evaluate the policies in two manners: i) the task success rate over 25 policy rollouts, and ii) the action serial numbers (right part of Fig. 3), i.e., the indexed numbers of the selected actions in the corresponding trajectory of the demonstration library, for each action of the rollout trajectories.

The task success rate for offline rollouts from human tactile demonstration is **40%** (10/25), whereas the success rate from Spacemouse-based teleoperation rollouts is only **12%** (3/25), and it from Hand-guided teleoperation is **28% (7/25)**. Then, in the bottom part of Fig. 3, we show the mean and variance of the action serial numbers of three successful rollout trajectories per policy. Human tactile demonstrations show a linear trend with minimal variance, while teleoperation policies display more nonlinearity and higher variance, especially during the insertion phase. This performance discrepancy arises because the majority of contacts occur during this phase, and the teleoperation systems which lack of human tactile feedback is not well-suited for capturing these contact-rich events. A similar conclusion can be drawn from the screenshots (top part of Fig. 3) for one successful rollout of all policies. Notably, the Human Tactile Demonstrations policy exhibits dynamic tilting for object insertion, which emulates the human's tactile-guided control strategy (see Fig. 1).

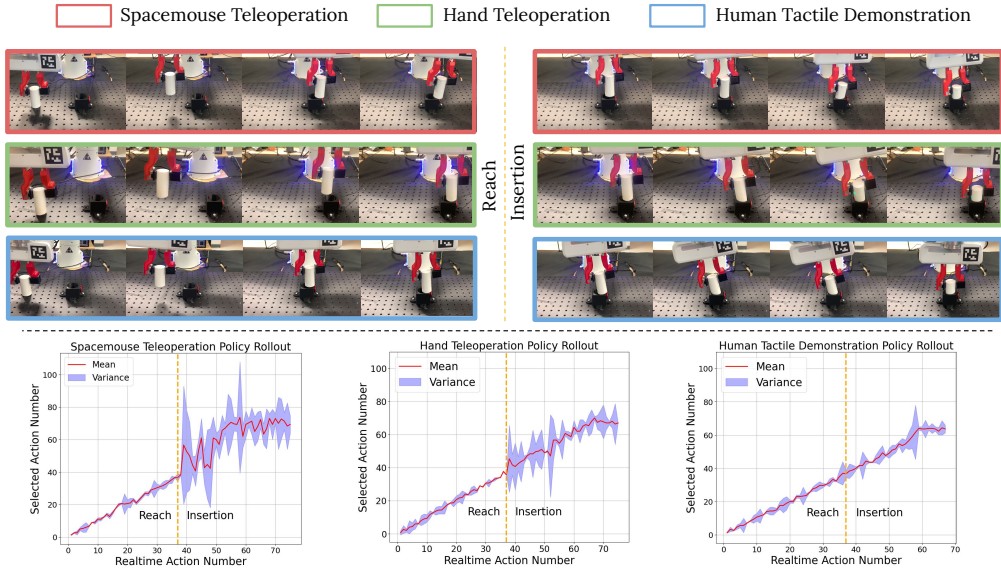

Figure 3: **Top:** Qualitative results for Spacemouse-based teleoperation, Hand-guided teleoperation, and Human Tactile Demonstration policies. **Bottom:** Visualization of the action serial numbers for three successful rollout trajectories generated by each policy. Solid red lines indicate mean trends and shaded areas show ± standard deviations. The left side of the dashed orange line is the Reach phase, and the right side is the Insertion phase.

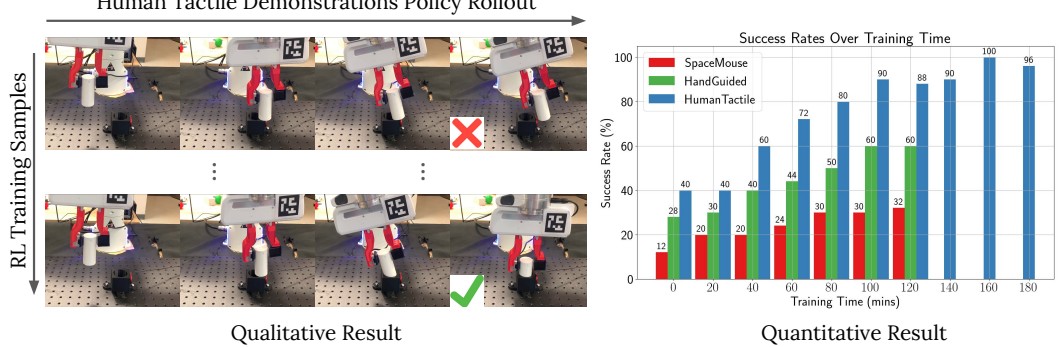

Figure 4: **Left:** Demonstrations of the online RL fine-tuning process, which further improves the task performance. **Right:** Quantitative task evaluations for offline policies learned from teleoperation demonstrations (SpaceMouse and HandGuided) and human tactile demonstrations (Human Tactile) during online RL fine-tuning show that Human Tactile significantly outperforms others in terms of task success rate and RL training efficiency.

Therefore, combined with the results in Sec. 4.1, MimicTouch not only efficiently collects human tactile demonstrations, but also enables effective policy learning using these demonstrations.

### 4.3 Online Policy Improvement and Generalization to New Settings

In this section, we evaluate the final policy trained through online RL. To ensure the robustness of the online policy, we perform domain randomization on the robot's starting state so that the initial object-hole contact is located differently. For the policy update, at each iteration, we use five newly collected trajectories along with another five randomly selected trajectories from the replay buffer.

**Online RL fine-tuning significantly and efficiently improves task performance.** We evaluate the trained policy every 20 minutes, approximately after every 13 RL epochs. After each hour, we compute the task success rates over 25 policy rollouts, in alignment with the offline policy evaluation. For other time instances, we only compute the task success rates over 10 policy rollouts to minimize sensor wear. The evaluation results are shown in Fig. 4, and we can observe that the policy learned with human tactile demonstrations can reach **96%** (24/25) task success rate in **3 hours**. Besides, it can reach 88% (22/25) task success rate after 2-hour RL fine-tuning, which is significantly more

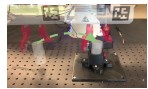 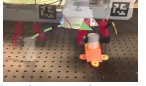 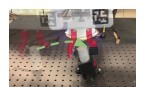 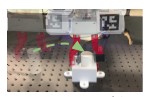 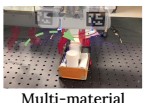 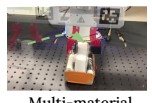 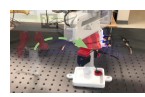

| Shifting 0.8 cm | Tilting 10 degree | Tilting 20 degree | Two-stage Packing | Multi-material Packing (Soft) | Multi-material Packing (Rigid) | Assembly |

Figure 5: Setup of zero-shot generalization tasks.

training efficient than the policies learned from both teleoperation demonstrations, which could only achieve 32% (8/25) and 60% (15/25) task success rates at that time, respectively. This result supports the effectiveness of online RL fine-tuning, as it allows the robot to further interact with the task environment. Moreover, it once again highlights the importance of using human tactile demonstrations since the offline policy learned from teleoperation demonstration exhibit significant training inefficiency. Finally, we compare the action outputs of the offline and online components of the MimicTouch final policy in Appendix. K to measure each one's contribution.

**MimicTouch policy exhibits superior zero-shot generalization capability.** We evaluate the zero-shot generalizability of the MimicTouch final policy. We consider the following generalization settings: i). *Shifting Positions (Shift)*: an insertion task with the hole shifting for 0.8cm in either $\pm x$ or $\pm y$ directions, ii). *Tilting Angles (10° and 20°)*: an insertion task with the hole angle tilted for 10° or 20°, iii). *Two-stage Dense Packing (Two-stage)*: a two-stage dense insertion task requires the robot to perform two consecutive alignments, iv). *Multi-material Dense Packing (Rigid and Soft)*: an insertion task where the box contains multiple objects, such as a pen, tissues, or a sponge, and v) *Furniture Assembly (Assem(I)) and Assem)* [53, 54]: an insertion task requires the robot to insert and thread the leg into a small hole of a table. Each setting is depicted in Fig. 5. To demonstrate the complexity of these tasks, we introduce a baseline: *Openloop Policy*, where we collect five successful insertion trajectories from the initial setting and execute each of these trajectories five times for each of the tasks. See Appendix. L for the details of each task and the policy evaluation process.

| Policy | Shift | 10° | 20° | Two-stage | Rigid | Soft | Assem (I) | Assem |
|---|---|---|---|---|---|---|---|---|
| *Openloop* | 24/40 | 14/25 | 10/25 | 13/25 | 13/25 | 9/25 | 8/25 | 3/25 |
| MimicTouch | **37/40** | **23/25** | **20/25** | **22/25** | **20/25** | **16/25** | **19/25** | **13/25** |

Table 2: Task success rate for each generalization task.

We report the task success rate of both policies in Table. 2. Based on these results, we can observe that the MimicTouch policy can significantly outperform all the openloop policies in all those generalization tasks. Also, since the *Openloop* policy has lower success rates in all the generalization tasks, it indicates the robustness of the MimicTouch policy in different challenging generalization domains. We also include the qualitative results and detailed analysis in Appendix. M.

## 5 Limitation and Future Work

MimicTouch pioneers the pathway to learning human tactile-guided control strategies from human tactile demonstrations. However, it still has several limitations for future improvements. Firstly, MimicTouch still requires several hours to refine the policy for addressing the embodiment gap. We will explore a better representation learning method to reduce the gap between humans and robots. Secondly, MimicTouch is task-specific and can not directly generalize human's tactile-guided control strategy to other tasks. One potential solution is to learn a generalizable tactile-based dynamic model for different tasks. Thirdly, the method of learning to perform contact-rich manipulation tasks from human tactile demonstrations could be extended to other robot tasks, such as dexterous manipulation, bimanual manipulation, and soft object manipulation.

## 6 Conclusion

We presented MimicTouch, a multi-modal imitation learning framework that: i) enables humans to perform tactile demonstrations with their hands without a robot in the loop, ii) learns from such demonstrations and safely transfers to robot with non-parametric imitation learning, and iii) improves the policy performance with residual-based online RL to bridge the human-robot embodiment gap. We show that MimicTouch enables high-throughput data collection and achieves high success rate and generalization across a wide range of two-piece insertion and assembly tasks.

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

# Appendices

## A   Data Collection

In this section, we show the novel data collection system in Fig. 6.

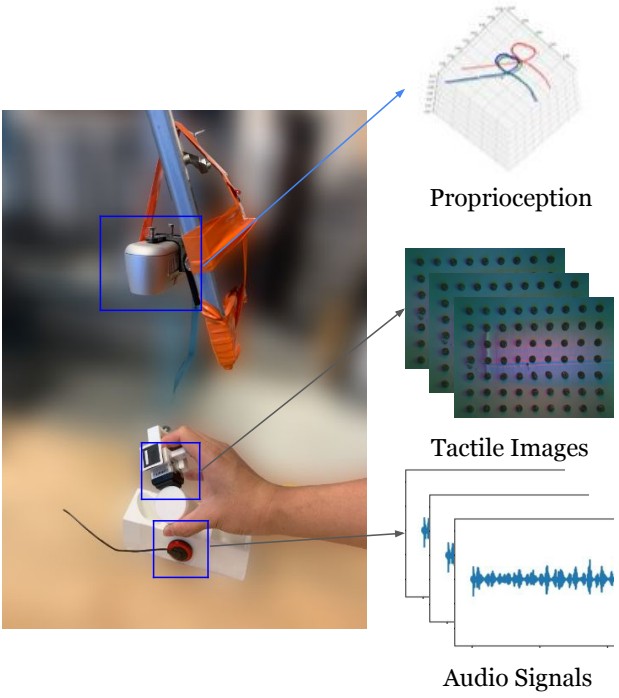

Proprioception

Tactile Images

Audio Signals

Figure 6: The human tactile demonstrations collection system.

## B   Representation Learning

**Data Collection** For Representation Learning, we collect a large task-specific dataset that contains 7657 tactile images and 1000 audio sequences. We collect 100 trajectories, each of them approximately five seconds long and containing around 70 tactile images and 10 audio sequences. These trajectories contain various data qualities, which include successful cases, failure cases, and sub-optimal cases. In detail, successful cases refer to the cases that human finishes the task with smooth trajectories; failure cases mean that human did not insert the object successfully or used more than five seconds to finish this task; sub-optimal cases mean that human used unnecessary motions to finish the insertion task.

**BYOL** BYOL [49] generates two augmented views, $v \triangleq t(x)$ and $v' \triangleq t'(x)$, from a given $x$ by applying image augmentations $t \sim \mathcal{T}$ and $t' \sim \mathcal{T}'$ respectively, where $\mathcal{T}$ and $\mathcal{T}'$ represent distinct augmentation distributions. The architecture of BYOL comprises a primary encoder $f_\theta$ and a target encoder $f_\xi$, where the latter being an exponential moving average of the former. Given the augmented views $v$ and $v'$, they are processed to yield representations $y$ and $y'$. These representations are subsequently transformed by projectors $g_\theta$ and $g_\xi$ to produce higher-dimensional vectors $z$ and $z'$. The primary encoder and its associated projector are designed to predict the output from the target projector, resulting in $q_\theta(z_\theta)$ and $sg(z'_\xi)$. The model's output consists of $l_2$-normalized versions of these predictions, which are trained using a similarity loss function. Post-training, the encoder $f_\theta$ is utilized for feature extraction from observations.

To utilize BYOL in tactile images, we scale the tactile image up to 256x256 to work with standard image encoders. We use the ResNet [55] architecture, also starting with pre-trained weights. Unlike SSL (self-supervised learning) techniques used in visual images, we only apply the Gaussian blur and small center-resized crop augmentations, since other augmentations such as color jitter and grayscale would violate the assumption that augmentations do not change the tactile signal significantly. For each input, the trained model will generate a $1 \times 2048$ representation vector.

**Audio Representation Learning** BYOL-A [50] is an extended version of BYOL to audio representation learning, processing log-scaled mel-spectrograms through a specialized augmentation module. To utilize BYOL-A in our audio data, we down-sampled signals from 44.1kHz to 16kHz, with a window size of 64 ms, a hop size of 10 ms, and mel-spaced frequency bins F = 64 in the range 60–7,800 Hz. Then, the Pre-Normalization step stabilizes the input audio for subsequent augmentations. Once normalized, the Mixup step introduces contrasts in the audio's background, defined by the log-mixup-exp formula:

$$\tilde{x}_i = \log((1 - \lambda)\exp(x_i) + \lambda\exp(x_k))$$

where $x_k$ is a mixing counterpart and $\lambda$ is a ratio from a uniform distribution. The next one is the RRC block, an augmentation technique, that captures content details and emulates pitch shifts and time stretches. For each input, the trained model will generate a $1 \times 2048$ representation vector.

## C  Data Pre-processing

### C.1  Sensor Data Alignment

Each sensor operates at different frequency: i) RealSense operates at 60 Hz with a resolution of 320x240 pixels, ii) GelSight Mini streams tactile images at 15 Hz with 400x300 pixel resolution, and iii) HOYUJI TD-11 piezo-electric contact microphone has a 44.1kHz sampling rate, and the audio data is segmented at 2Hz, meaning each audio input to our framework is a 0.5s sequence containing 22,050 audio signals.

Therefore, to ensure synchronization across our sensors, we first address the disparate sampling rates of the fingertip poses, tactile images, and audio sequences, which are 60Hz, 15Hz, and 2Hz, respectively. Specifically, we downsample the fingertip poses to 15 Hz. For the audio data, instead of collecting entirely new 0.5-second segments, we record the extended audio signals at intervals of every 0.07 seconds. As a result, each 0.5s segment has a new-collected 0.07s interval and an old overlapped 0.43s segment in the past, which results in an overlap of 0.43 seconds between consecutive audio segments. Therefore, all sensor inputs are sampled at 15Hz.

### C.2  Calibration and Filtering of Fingertip Poses

The 6D human fingertip poses extracted from the AruCo marker include 3D positions along with rotation vectors. To use these fingertip poses as the end-effector's poses for robot policy learning, we need to address two problems: i). developing a calibration method to align the data collection system with the robot execution system, and ii). implementing a filtering method to generate smooth trajectories.

**Calibration** Given that data collection and robot experiments occur in disparate scenarios, it is crucial to align our human tactile data collection system with the physical robot system. Initially, we record the distance between the object (starting point) and the base (ending point) within the data collection system and replicate this setup in the robot environment. Following this, six equidistant positions between the starting and ending points are identified within both systems. The object is gripped at these predetermined positions using both hands and the robot's end-effector so that we can capture the corresponding poses. In this calibration process, the hand poses, denoted as the "Eye" in the calibration function, are referenced to the camera frame, while the end-effector poses, represented as the "Hand" in the calibration system, are referenced to the robot frame. Conclusively, we employ the calibrateHandEye function from OpenCV, using the six captured poses, to calibrate

---

**Algorithm 1** Online Residual Reinforcement Learning

---
1: **Input:** offline policy $\pi_o$, randomly initialized residual policy $\pi_r$
2: **Input:** step size sequences $\{\beta_t\}$, number of iterations $K$, Replay Buffer $D$
3: Initialize replay buffer $D$ with pre-collected data
4: **for** $k = 1$ to $K$ **do**
5:     Sample mini-batch $D_k$ from Replay Buffer $D$
6:     Obtain current trajectory $C_k$ by executing $\pi_o + \pi_r$
7:     To collect more data in $D$: $D \leftarrow D \cup C_k$
8:     Combine $D_k$ and $C_k$ to form batch $B_k$ for update
9:     **for all** $(s, a_i, r, s') \in B_k$ **do**
10:         $a_o \leftarrow \pi_o(s)$                                          $\triangleright$ Obtain offline action
11:         $a_r \leftarrow \pi_r(s, a_o)$                                   $\triangleright$ Obtain residual action
12:         $\hat{a} \leftarrow a_o + a_r$                                  $\triangleright$ Combine offline and residual actions
13:         $Q_r \leftarrow Q_r(s, a_o) + r + \gamma Q_r(s', \pi_o(s'))$
14:         $\pi_r \leftarrow \pi_r - \alpha \nabla_{\pi_r} L(\pi_r)$        $\triangleright$ Update residual policy with gradient step
15:     **end for**
16: **end for**
17: **Return:** Trained residual policy $\pi_r$

---

these two frames (camera frame and robot frame). It should be noted that during robot execution, we do not use a camera. Therefore, we cannot collect human fingertip trajectories in the camera frame and use inverse kinematics for robot execution, as is commonly done in vision-based policies.

**Filtering** Given the inherent noise and occasional outliers in the poses obtained from the RealSense and AruCo markers, it is imperative to implement post-processing techniques to ensure the quality and smoothness of the trajectories. For each pose sequence, outliers are detected by sorting the values of each delta transformation (i.e., the delta translations and the delta Euler angles). The Interquartile Range (IQR) method is employed to establish the upper and lower bounds, which are then used to identify outliers. The IQR is defined as: $\text{IQR} = Q_3 - Q_1$ where $Q_3$ and $Q_1$ are the third and first quartiles, respectively. Outliers are replaced using a median filter with a window size of 3. To enhance the temporal consistency of the estimated hand and object pose, a digital low-pass filter is applied to eliminate high-frequency noise. Specifically, the filter has a sampling frequency of 5Hz and a cutoff frequency of 2Hz. The low-pass filter can be represented as: $H(f) = \frac{1}{1 + (\frac{f}{f_c})^2}$ where $f$ is the sampling frequency and $f_c$ is the cutoff frequency.

## D   Details of RL training

**Training pipeline** The overall pseudo-code for RL Training is given in Alg. 1.

**Action limits** The residual RL policy is designed to make only slight adjustments to the offline IL policy. To ensure this, during RL fine-tuning, we constrain the action ranges of the residual policy to (-0.15cm, 0.15cm) for each x, y, z translation and (-3 degrees, 3 degrees) for each Euler angle.

**RL policy details** For the residual policy $\pi_r$ within our framework, we employ the Soft Actor-Critic (SAC) algorithm with an MLP architecture. The training strategy aims to effectively combine reinforcement learning principles with residual corrections, thus enhancing the overall performance of the system. The following formula represents the objective for training the residual policy:

$$\pi_r = \underset{\pi}{\text{argmax}} \left\{ \mathbb{E}_{(s,a) \sim D} \left[ Q(s, a + \pi(s)) - \alpha \log \pi(a|s) \right] \right\}$$

- $Q(s, a + \pi(s))$: The Q-value function, which estimates the value of executing the residual action $\pi(s)$ in addition to the base action $a$ in the state $s$. This represents the total action influenced by both the offline policy and the residual corrections suggested by $\pi_r$.

- $\mathbb{E}_{(s,a) \sim D}$: The expectation over state-action pairs sampled from the replay buffer $D$, which contains data from both past experiences and current new explorations.

- $\alpha \log \pi(a|s)$: The entropy regularization term for the policy $\pi$, which encourages exploration by penalizing the certainty of the policy's action selection. This term is crucial in SAC to ensure sufficient exploration and avoid premature convergence to suboptimal policies.

- $\pi(a|s)$: The policy network (MLP) outputs the probability distribution over actions given the state $s$, from which the action $a$ is sampled.

This formula ensures that the residual policy $\pi_r$ learns to adjust the actions generated by the offline policy by optimizing the SAC objective. It balances the maximization of expected returns (via Q-values) and the maintenance of behavioral diversity (via entropy regularization), allowing $\pi_r$ to adapt and refine actions based on real-time environmental feedback and historical data from the replay buffer.

**RL Reward Design** We will give a detailed explanation for each component in our reward design.

**Distance Reward:**
$$d = 1 - tanh(10.0 * ||distance||_2)$$
where the *distance* is between the current position of the gripper center and the target gripper center.

**Orientation Reward:**
$$o = 1 - tanh(7.5 * ||diff\_ori||_2)$$
where the $diff\_ori$ stands as the quaternion difference between the current gripper orientation and the target gripper orientation.

**Penalty for blocking**

$$c = \begin{cases} 0.2, & \text{successfully complete this action} \\ 0, & \text{cannot complete this action in 0.5s} \end{cases} \tag{1}$$

**Penalty for Slippery**

$$s = \begin{cases} 0.5, & ||y_t^i - y_t^{i-1}|| \geq 0.5 \\ 0, & \text{otherwise} \end{cases} \tag{2}$$

The $y_t^i$ and $y_t^{i-1}$ stands for the embeddings of the tactile images in the current step $i$ and the last step $i-1$.

**Overall Rewards**

$$R = \begin{cases} 1, & \text{if success} \\ \alpha D_{KL}(P\|Q) + \beta d + \gamma \cdot o - c - s, & \text{otherwise} \end{cases} \tag{3}$$

In Eqn. 3, $P$ is the executed trajectory, $Q$ is the expert trajectory, and $D_{KL}(P\|Q)$ is the KL Divergence between the executed trajectory and the expert trajectory, which makes the executed trajectories close to the expert trajectory, ensuring that the robot does not take risky behavior. In addition, $d, o, c, s$ are defined above separately. The setup of each weight: $\alpha = 0.5, \beta = 0.3, \gamma = 0.2$.

# E  Hand-guided Teleoperation Setting

As shown in Fig. 7, the human operator can directly control the robot end-effector. This method offers a more intuitive approach to data collection and has proven effective in manipulation tasks [10].

# F  Details of the inserted objects and the holes

In all the experiments, except for the furniture assembly, the object being inserted is a cylinder with a radius of 17.5 mm and a height of 80 mm. In both the shifting and tilting experiments, the only variable among the holes is the angle between the hole opening and the horizontal plane, which is

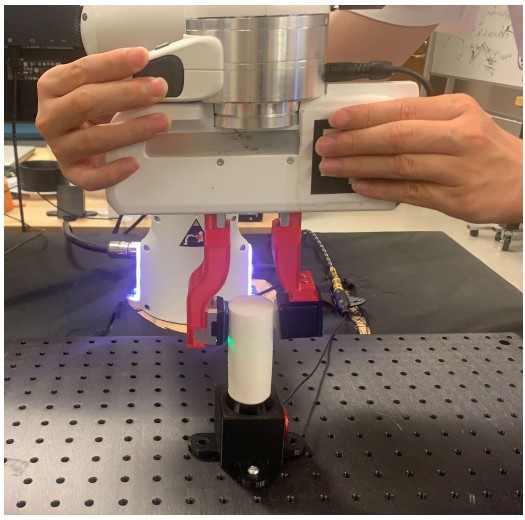

Figure 7: Hand-guided Teleoperation System.

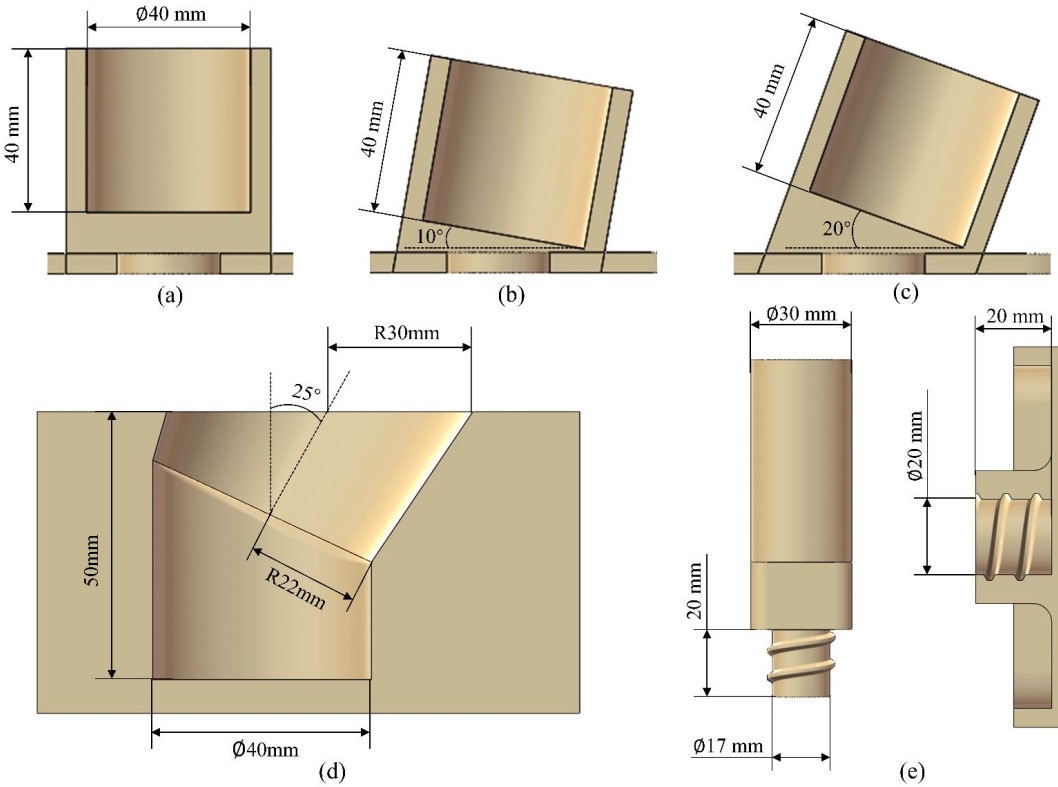

Figure 8: Detailed dimensions of the inserted objects and holes.

set to 0°, 10°, and 20°, respectively (see Fig. 8, parts (a), (b), and (c)). For all, the hole openings have a uniform radius of 20 mm and a depth of 40 mm.

In the two-stage packing experiment, the hole consists of two sections. The first section is a flared hole with a bottom radius of 22 mm, a top radius of 30 mm, and an angle of 25° relative to the horizontal plane. The second section is a cylindrical hole with a radius of 20 mm. The combined height of the two parts is 50 mm (see Fig. 8, part (d)).

In the last furniture assembly experiment, the threaded section of the inserted objects is a cylinder with a radius of 8.5 mm and height of 20 mm (see Fig. 8, left of parts (e)). It connects to the tabletop through a thread with a diameter of 17 mm and a pitch of 8 mm. To facilitate tightening, the radius of the threaded hole in the tabletop is 10 mm (see right of part (e)).

## G  Emulate the Physical Environment for Policy Evaluation

3To emulate the physical robot environment, we introduce random noise to those 10 unseen data sequences. The robot state space input undergoes a random position noise within the range $[-3\text{cm}, +3\text{cm}]$ for each axis. Gaussian noise, denoted as $\mathcal{N}(0, \sigma)$, is added to both the tactile image and audio signal. In this notation, $\mathcal{N}(0, \sigma)$ signifies a Gaussian distribution with a mean of 0 and a standard deviation of $\sigma$. For tactile images, the noise affects pixel values in the range $[0, 255]$, while for audio data, it impacts signal values in the range $[0, 1]$. Given their distinct ranges, we apply Gaussian noise with standard deviations of $\sigma = 100$ for tactile images and $\sigma = 0.4$ for audio data.

## H  MSE Loss

For calculating the MSE Loss between two action sequences, we need to normalize the actions' translation vectors and rotation vectors since they have different scales. Specifically, we use min-max normalization on both the translation vectors and rotation vectors, where the max vector and min vector are selected from the training set. As a result, translation vectors and rotation vectors will have the same scale for calculating the MSE Loss. The formula is shown as:

$$\text{MSE} = \frac{1}{n} \sum_{i=1}^{n} (y_i - \hat{y}_i)^2$$

Where: $y_i$ represents the ground truth normalized action, $\hat{y}_i$ represents the generated normalized action, and $n$ is the number of all action steps.

## I  Failure Examples of the MULSA Policy

We provide three failure examples to better illustrate why the MULSA policy performs worse. As shown in Fig. 9, when the robot makes contact at the wrong position or fails to make contact, it tends to drift further away from the hole. This occurs due to covariant shift and compounding errors [44], leading the policy to continue to select erratic actions, ultimately resulting in task failures. Based on these observations, we believe the MULSA policy is less suitable for real-world RL fine-tuning.

## J  Ablation Study: Do Multi-Modal Tactile Feedback Improve the Task Performance?

In this section, we evaluate the performance of our multi-modal tactile embeddings. We consider the following baselines: i). *MimicTouch w/o T & A*: MimicTouch without tactile or audio embeddings, ii). *MimicTouch (T)*: MimicTouch incorporating only tactile embeddings, iii). *MimicTouch (A)*: MimicTouch incorporating only audio embeddings., and iv). *MimicTouch (T + A, Ours)*: MimicTouch incorporating both tactile and audio embeddings. We evaluate the policy performance in terms of the MSE losses over the test sets (see Sec. 4.2.1) and task success rates over 25 policy rollouts. In addition, we visualize the impact of each sensor modality during policy execution. Specifically, we plot the normalized distance between the query feature and the selected key feature for each sensor input. A larger distance means that the corresponding sensor modality contributes more in selecting the key feature from the demonstration library.

From Table. 3, we observe that without using both tactile images and audio signals, the MSE loss (task success rate) is 0.62 (4%), which is significantly higher (lower) than the others. By incorporating both tactile and audio feedback, the MSE loss can be as low as 0.21, and more importantly, the task success rate can reach 40%.

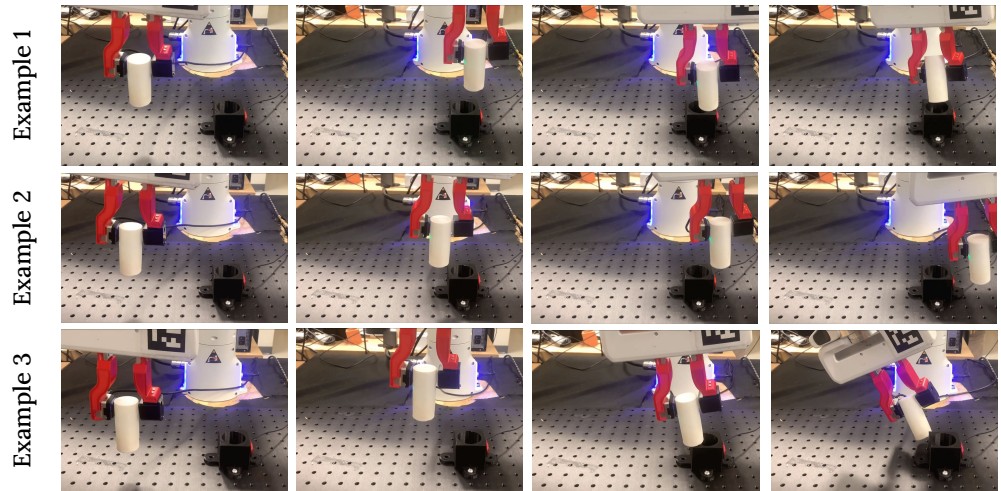

Figure 9: Three failure rollouts of the MULSA policy. For all of them, the robot drifts away from the hole, resulting in complete task failures.

| Models | MimicTouch w/o T & A | MimicTouch (T) | MimicTouch (A) | MimicTouch (T + A) |
|---|---|---|---|---|
| MSE Loss | 0.62 | 0.38 | 0.48 | **0.21** |
| Success Rate | 4% (1/25) | 24% (6/25) | 16% (4/25) | **40% (10/25)** |

Table 3: MSE Loss over test sets and Task success rates of 25 policy rollouts.

As shown in Fig. 10, tactile and audio inputs start to play important components during the Insertion phase, when most contacts occur. We also have qualitative results shown in the Fig. 11. According to those results, we can find that a lack of tactile feedback easily leads to incorrect motion when contact appears, whereas the lack of audio feedback easily leads to an inability to detect external collisions.

Therefore, we can conclude that the multi-modal tactile feedback is crucial for the success of contact-rich insertion tasks.

## K Comparison of Action Magnitudes Between Offline IL Policy and Residual RL Policy

To quantitatively evaluate the contributions of both the IL and RL components of the final policy, we compare the action outputs (i.e., the $6D$ delta pose) of each across three successful rollout trajectories. In the left plot of Fig. 12, we show the average distances and in the right plot, we show the average rotation angles. From these results, we observe that the IL policy's action outputs account for a higher proportion of the final output, indicating that the IL policy is overall more critical. This underscores the importance of collecting Human Tactile Demonstrations and processing them using the NN-based method. However, it is also important to note that the smaller contributions from the RL policy are also necessary for the successful execution of the fine-grained manipulation tasks.

## L Generalization Setting

In this section, we describe the setting of each generalization task.

**Shifting Positions:** In this generalization task, we shift the hole positions for 0.8 cm in either $\pm x$ or $\pm y$ to test the generalization ability for finishing the insertion task with under varied alignment conditions.

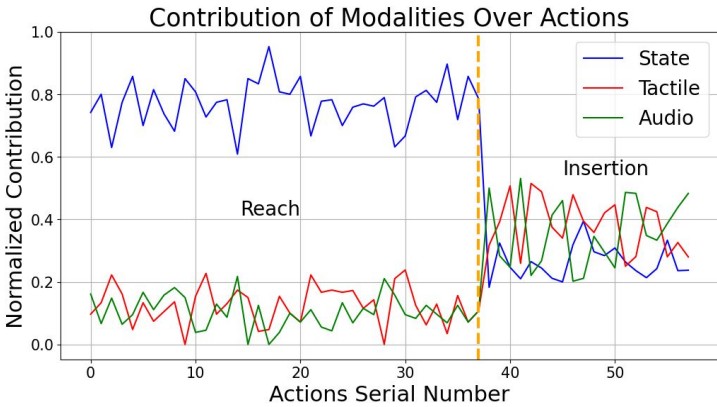

Figure 10: Visualization of the impact of each sensor modality during policy execution. The left side of the dashed orange line is Reach Phase, and the right side is Insertion Phase.

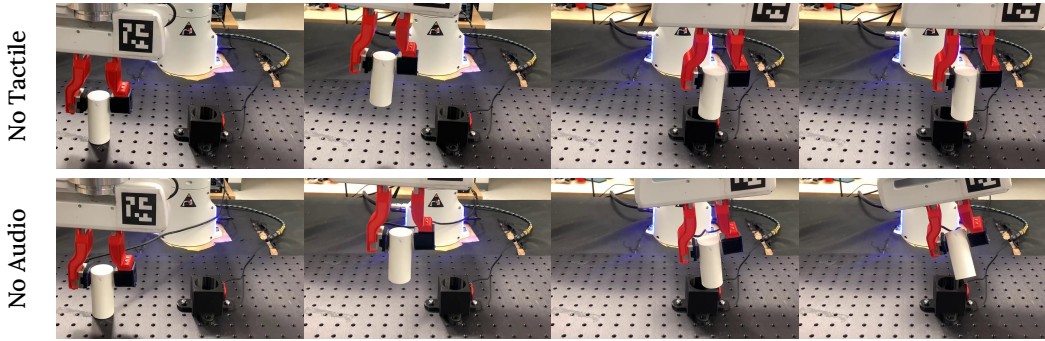

Figure 11: Policy rollouts of some failure examples with only tactile feedback or only audio feedback.

**Tilting Angles:** In this generalization task, we tilted the hole angle for 10 degrees or 20 degrees to test the generalization ability for finishing the insertion task with different contact positions.

**Two-stage Dense Packing:** We introduce the two-stage dense packing task, which requires the robot to perform two consecutive alignment adjustments to complete the dense packing process. Each hole will challenge the robot's ability to adjust its alignment according to tactile feedback efficiently.

**Multi-material Dense Packing:** In this generalization task, the robot is required to insert the cylinder into the hole which contains multiple objects: pen, tissue, and sponge. This setting has rigid objects (pen) and deformable soft objects (tissue and sponge) with different materials and shapes, which challenge the robot's ability to accomplish the task with different tactile feedback from different materials.

**Furniture Assembly:** In this generalization task, the robot is required to insert the cylinder into a small hole in a table for screwing. This task will test two aspects of our policy: whether the robot can insert the object into a smaller hole, and whether the robot can adjust it to a position, that is deep enough to be skewed by a human-defined simple script (to rotate the end-effector for 120°), based on the tactile feedback from the threads in the hole.

**Policy Evaluation Process:** We evaluate the policy performance in those five different generalization settings for both MimicTouch final policy and the *Openloop* Policy. For *Shifting Positions*, we rollout the policies 10 times in each direction of $\pm x$ or $\pm y$, resulting in a total of 40 evaluations. For *Tilting Angles*, we rollout the policies 25 times for both tilting directions in $\pm x$, resulting in a total of 50 evaluations for the $10°$ tilting and $20°$ tilting, respectively. For *Two-stage Dense Packing*, we rollout the policies 25 times. For *Multi-material Dense Packing*, we rollout the policies 25 times on

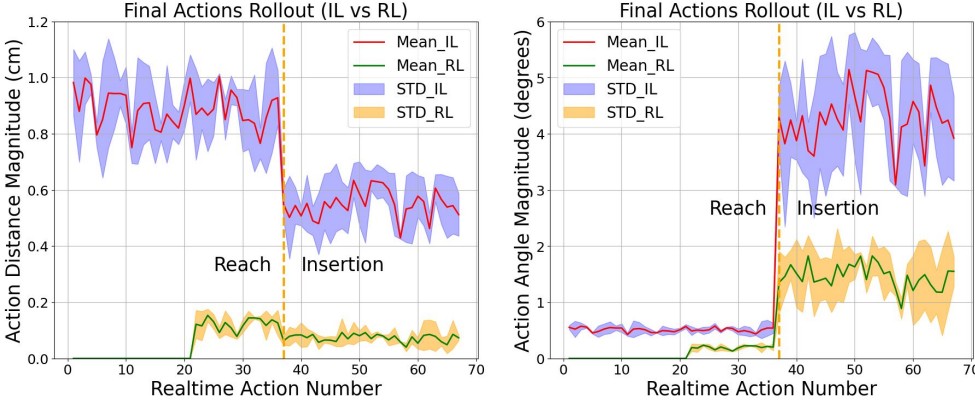

Figure 12: In the left, we show the average distances and in the right, we show the average rotation angles. We can observe that the IL policy's action outputs account for a higher proportion of the final output.

both rigid object "pen" (Rigid in the table) and deformable soft objects "tissue and sponge" (Soft in the table). For *Furniture Assembly* (Assem in the table), we rollout the policies 25 times. This task has two sub-evaluation metrics: insertion (Assem (I) in the table) and adjustment (Assem (A) in the table). Notably, the success rate for Assem (I) is the success rate for the number of attempts (25), and the success rate for Assem (A) is the success rate when the insertion is successful.

## M    Generalization Results

In this section, we summarize the zero-shot generalization results based on the quantitative results shown in Table. 2, and the qualitative results shown in Fig. 13.

- MimicTouch policy can zero-shot transferring to insertion tasks with different contact positions, tilted angles, and even different sizes of holes (Aseembly (Insertion)).

- For the Two-stage Dense Packing, MimicTouch policy displays its robustness to adjust according to multiple stages of contact information according to the quantitative result and the video. This shows that our model can make continuous and correct adjustments based on the continuously varied contact information.

- MimicTouch policy also shows its power in the multi-material task domains. Due to different materials in the environment, the sensor feedback will be different from the training environment. In this case, our policy still has great performance on other rigid objects (pen). For the deformable soft object (tissue and sponge), The success rate is a bit lower because of two challenging issues: i). it's hard to get audio feedback for contact with soft objects, ii). sponge sometimes is too soft to get tactile feedback. With those issues, our policy still gets 64% success rate. Moreover, the qualitative result from the video shows impressive performance in adjusting continuously according to the deformation of the tissue.

- In the assembly task, MimicTouch policy not only can insert the object into a small hole but also can adjust the object to a correct pose and insert it to a deep-enough position according to the tactile feedback from the contact between screw threads. This allows us to use a very simple script (to rotate the end-effector for $120°$) to solve the assembly task.

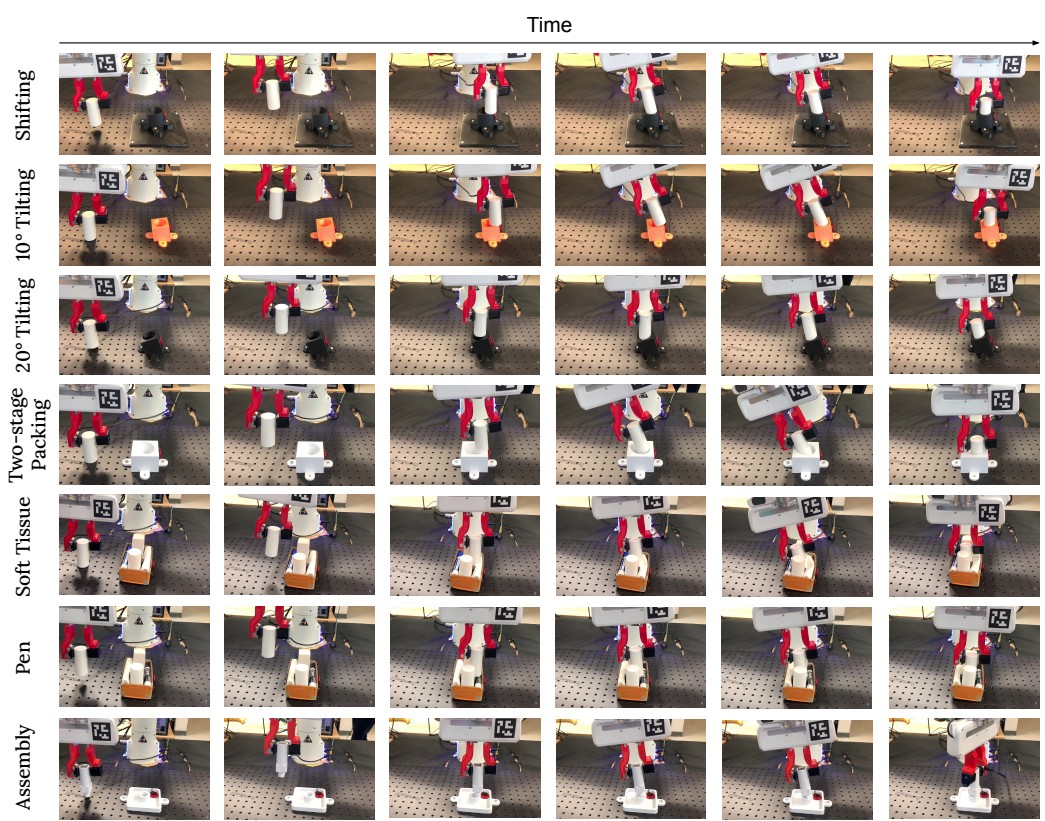

Figure 13: Task setup and qualitative results for zero-shot generalization tasks

