# OpenReview forum: "MimicTouch: Leveraging Multi-modal Human Tactile Demonstrations for Contact-rich Manipulation"
_robot-learning.org/CoRL/2024/Conference — CoRL 2024_

### Official Review · Reviewer_VHmX · 2024-07-05
**Well written although lacking proof why a 3-stage learning process (SSL->IL->RL) is needed**

**Originality:** 3
**Technical Quality:** 2
**Clarity Of Presentation:** 3
**Potential Impact:** 2
**Recommendation:** 3
**Confidence:** 5

**Review:**

The paper is overall clearly written and easy to follow. Structure-wise, I’d recommend the authors to rearrange the paper a bit as much useful information is hidden in the appendix, which is hard to grasp. Some evaluation sections can be cut shorter or moved to the appendix. Overall, although the introduced method is novel and the results are interesting, I’m not convinced that the 3-stage learning process (SSL->IL->RL) is necessary. I don’t see why a modern IL policy or a e2e RL policy cannot perform the experimented task, which is a simple peg-in-hole task. An ablation study on this pipeline will be helpful.

In summary, I don’t think the paper in its current state is suitable for publication at CoRL mainly due to the lack of proof of the effectiveness of the proposed pipeline v.s SOTAs and the lack of more complicated tasks.


Fig. 1: since this figure appears right underneath the title and it is likely the first thing that readers will read, I’d suggest providing more explanations regarding what exactly those zero-shot generalizations mean, i.e. are they novel tasks? what are the differences between those tasks and the ones shown on the second row?

Sec 3.2: for non-audio experts, what does “we segment the audio data at 2Hz” mean? Sampling audio signals at 2Hz sounds overly too low. Or did the authors mean 2s?

I would encourage the authors to spend a bit more ink on Sec 3.2 to explain the methods and architecture used in the work. Although more details are indeed provided in the appendix, I think those are integrated parts of the paper and it is worth a few sentences to provide an overview.

Sec 3.3: If SAC learns a residual policy pi_r and the final action output will be a_i + a_r, shouldn’t pi_r also be a function of a_i? In my mind it would be better if pi_r is a function of both the state vector s and the IL policy output a_i, because if pi_r only learns a residual action from the state vector, I’d imagine pi_r implicitly also need to learn a_i to know how to compensate for it. Please justify.

In general, I’d encourage the authors to talk more about the specific methods (e.g. moving some stuff from the appendix to the main paper) and maybe cutting some subsections of Sec 4 short, because it’s quite disruptive to move back and forth between the main sections and the appendix. For example, I don’t think Sec 4.1 is super necessary considering Table 1 is pretty self explanatory. A few sentences should be sufficient.

I’m not very convinced by the experiments in 4.2.1 that compare the performance between MULSA and VINN - for manipulation I think task success rate in real-world rollouts is (unfortunately) the only reliable way to compare policies. Higher MSE loss doesn’t necessarily mean worse policy.

Interesting findings in 4.2.2. However, was it because the tele-op data’s quality wasn’t high enough? Is that something that can be fixed with more intuitive ways of teleoperation e.g. VR or mo-cap?

**Quality Of The Limitations Section:**

3

**Questions For Rebuttal:**

1. Since there is a RL process at the end to learn a residual policy, how important are the representation learning process and the IL process? Can the proposed method actually beat the performance of e2e RL or e2e of a modern IL policy (e.g. diffusion policy or mulsa)?
2. Why doesn’t the residual policy pi_r also take a_i (the action output from IL) as an input?
3. Not really a question, but I’d encourage the authors to rearrange the paper a bit and move some architecture/method explanations from the appendix to the main paper.

**Robotics Focus:**

4

**Summary Of Paper:**

This paper introduces a novel multi-modal manipulation learning pipeline from human demonstrations. In this pipeline, a human operator collects demonstrations with pose, tactile, and acoustic signals, then a BYOL-based SSL architecture is used to learn a representation, from which a IL policy is trained. At the end a RL policy learns a residual policy to compensate for the IL policy.

**Summary Of Recommendation:**

In summary, I don’t think the paper in its current state is suitable for publication at CoRL mainly due to the lack of proof of the effectiveness of the proposed pipeline v.s SOTAs and the lack of more complicated tasks.

---

### Official Review · Reviewer_4Bsg · 2024-07-19
**Reviewer comment**

**Originality:** 4
**Technical Quality:** 4
**Clarity Of Presentation:** 4
**Potential Impact:** 4
**Recommendation:** 3
**Confidence:** 4

**Review:**

This study presents learning from human tactile demonstration for contact-rich manipulation. The end-effector pose and tactile and audio information are collected during human demonstrations. Given this information, this study employs a prior imitation learning method [12] and proposes a residual RL approach to align human demonstrations and facilitate completing the task.
This study performed real-robot experiments. The result revealed that human tactile demonstrations provided better quality data than teleoperated demonstrations. Also, the residual RL improved the success rates and demonstrated generalization.

Overall, this paper is excellently written. Human tactile demonstrations should be useful for contact-rich tasks. I also experienced difficulty in teleoperation for contact-rich tasks. The figures and the video are also beautiful.

I would recommend accepting this paper for the conference. Please find my comments.

a) The residual approach is a good idea, but I'm a little bit concerned about whether it is too dominant.
Could this study plot the actions generated from the residual and imitation policies before and after fine-tuning? I want to see how much the plot contributes to the residual policy.
Also, can you train only the residual policy and see the performance with only it?
If the residual policy is too strong, the imitation learning may not be useful, resulting in weak novelty.

b) Adding more detailed information on VINN [12] would help readers—for example, the difference between the proposed method and VINN.

c) This study used a space mouse as a teleoperation method. If a VR controller is used, can the performance be improved? If possible, please try it.

d) Please provide more detailed information about the inserted objects (e.g., the dimension and the tolerance between the peg and hole).

e) Please describe the failure cases of the proposed methods. This will benefit the follow-up studies.

**Quality Of The Limitations Section:**

3

**Questions For Rebuttal:**

Please address the above comments, mainly focusing on comment a).

**Robotics Focus:**

4

**Summary Of Paper:**

This study presents learning from human tactile demonstrations for contact-rich manipulation.

**Summary Of Recommendation:**

Overall, this study is suitable for this conference. I would recommend weak accept. However, several points should be addressed. After rebuttal comment: Thank you very much for the detailed rebuttal. The additional experiments significantly improved the quality.  Although I would like to see direct comparisons between the proposed and pure RL methods in this environment, I understand the difference. I believe this paper is suitable for this conference. I would like to maintain the score.

---

### Official Review · Reviewer_W9Dy · 2024-07-20
**CORL 2024 Submission 387 Review**

**Originality:** 4
**Technical Quality:** 3
**Clarity Of Presentation:** 3
**Potential Impact:** 3
**Recommendation:** 3
**Confidence:** 5

**Review:**

The overall framework of learning multi-modal robot policies directly from human demonstrations is impressive and has the potential to enable simple data scaling for robots in the future. However, I believe the paper lacks sufficient robot experiments to support the decisions made throughout their framework and could benefit from some improvements and more explanations in a few parts.

The paper is mostly easy to understand, and the figures are helpful, but it could use more clarification in some sections. See below for recommended improvements.

The major contribution of the paper is the attachment design and the integration of several advancements in the robot learning community (such as residual RL, NN-based action retrieval, and self-supervision for tactile and audio) to create a framework for learning robot policies from human demonstrations.

**Strengths:**

- The overall implementation for learning multi-modal robot policies from human demonstrations is a valuable contribution.
- I found the comparison of this framework to teleoperation insightful. It is nice to see how both policies learned and the time required for collecting demonstrations improve with human demo collection. I found Figure 3 helpful.
- I enjoyed the generalization experiments. Although I believe a few more baselines for Table 2 would improve the insightfulness, it’s interesting to see that robot policies from human demonstrations can generalize.

**Weaknesses:**

- I believe the biggest weakness of the paper is the lack of robot experiments. The authors experiment only with a single task, and for many of the ablations, they only use MSE to compare their method to its baselines.
- Calibration seems a bit intricate in this work, which could cause difficulties in using this framework. From what I understand, for each new task, they need to measure the distance of the object to the robot base, collect 6 poses on the robot and the human hand, and then find the transform between these poses. This requires new calibration for each task and could present many problems. Please refer below for recommendations for calibration.
- I think the paper needs a few modifications to clarify some parts, including offline policy, generalization experiments, calibration, and reward calculation. Please refer below for a few recommendations.

**Quality Of The Limitations Section:**

3

**Questions For Rebuttal:**

Here is the list of additional experiments and recommendations that I believe could improve the paper:

**Calibration Related Improvements / Recommendations:**

- I recommend that the authors calibrate the robot’s base only once with respect to the camera and use that transformation for inverse kinematics. Is there a specific reason why they didn’t do this?

**Residual RL Related Improvements / Recommendations:**

- Could you explain how you set the action space during residual learning? Do you learn the residual policy for all dimensions of the action space, and do you have any action limits for your residual policy?
- During reward calculation, the authors mention using KL divergence to align the human expert and the robot data. What exactly is the input of this KL divergence? Do they include only the proprioception, or do they include the tactile and the audio as well?
- A small explanation of the RL learning in the main paper for these questions could be useful.

**Experiments Related Improvements / Recommendations:**

- For Section 4.1 Human Tactile Demonstration Collection System: Could you please reiterate how you calculate the success rate here? Are you only running trajectory replay of the demonstration to see if that succeeds, or are you counting the number of trials during data collection? If it is trajectory replay, could you explain why the teleoperated demonstrations fail? If they succeeded during data collection, why would they fail in the open-loop trajectory?
- The authors used MSE on the test set to compare their method with the baselines, both for different modalities and different offline policies. I find this evaluation somewhat deficient. For offline policies, NN-based action retrieval compared to a parametric policy would, of course, give lower MSE since it outputs very similar actions to those in the test set. However, a parametric offline policy could perform better in situations that require different actions than those in the demonstration set. I recommend that the authors run real robot experiments for both Section 4.2.1 (offline policy learning) and Table 3 (modality ablation).

**Generalization Related Improvements / Recommendations:**

- Could you explain the generalization experiments in more detail? Are you using the collected 5 demonstrations for the MimicTouch in action retrieval as well, or are you using the old demonstrations for the offline policy and training a residual policy on top?
- Could you include NN-based action retrieval on the new demos and NN-based action retrieval on the old demos with the residual RL as baselines for Table 2? This could help understand how much the residual RL improves generalization and how much better the NN-based policy performs atop open-loop.

**Syntax / Writing Related Improvements / Recommendations:**

- Instead of using percentages for the success rates of the policy deployments, I recommend using <# of successful runs> / <# of evaluation runs>. It makes it easier to read and comment on.
- For Figure 7, Contribution of Modalities, the authors mention that as the distance of the modality is higher, it has a higher contribution. I didn’t fully understand this part. Wouldn’t it be that as the distance of the modality is lower to the nearest neighbor, it has a higher contribution? Could the authors explain this, please?
- The authors refer to their offline policy as VINN. However, VINN uses interpolation between a set of K nearest neighbors as the final action, but from what I understand, MimicTouch only uses the nearest neighbor's action without any interpolation. If that’s the case, I recommend adding this small difference and not referring to their offline policy as VINN, perhaps using NN-based action retrieval or something similar.

**Robotics Focus:**

4

**Summary Of Paper:**

The paper introduces a framework to collect multi-modal human demonstrations and use them for robot policies. The embodiment involves a robotic arm with a gripper, and optical-tactile and audio sensors attached to it. Humans can wear an attachment with ArUco markers and the same sensors attached, allowing them to collect demonstrations using their fingers as a gripper. Calibration between the robot and the human hand is done using a set of poses collected by both the human and the robot hand. After data collection, they train tactile and audio representations using self-supervision and use nearest neighbor-based action retrieval as an offline policy. They fine-tune their offline policy by learning an online residual policy added atop the offline policy. They evaluate their method on a peg insertion task and showcase its generalization capabilities in various settings for that task.  The major novel contribution of the paper includes a framework to collect human audio-tactile demonstrations, including the design of an attachment for use and deploying policies learned on a gripper. There is no significant originality in the policy learning aspect of the framework.

**Summary Of Recommendation:**

Even though I find this work and overall framework valuable to share with the community, I believe there are significant robot experiments to be made before fully accepting the paper. Post Rebuttal Edit: I believe after the rebuttal, the paper is now suitable for the conference.

---

### Author Rebuttal · Authors · 2024-08-12

We thank all Reviewrs' careful review and suggestions. We have added new experiments and explanations to address each of your concerns and have uploaded the latest version of the paper accordingly, with all changes highlighted in blue.

---

### Decision · Program_Chairs · 2024-09-04

**Decision:**

Accept

**Comment:**

All reviewers concur that the paper should be accepted. Please incorporate reviewer feedback in the camera ready.